

# The voluntary control of piloerection

James A.J. Heathers[1,*], Kirill Fayn[2,*], Paul J. Silvia[3], Niko Tiliopoulos[4] and Matthew S. Goodwin[1]

[1] Bouve College of Health Sciences, Northeastern University, Boston, United States of America
[2] Department of Quantitative Psychology and Individual Differences, KU Leuven, Leuven, Belgium
[3] Department of Psychology, University of North Carolina at Greensboro, Greensboro, United States of America
[4] Department of Psychology, University of Sydney, Sydney, Australia
[*] These authors contributed equally to this work.

## ABSTRACT

Autonomic nervous systems in the human body are named for their operation outside of conscious control. One rare exception is voluntarily generated piloerection (VGP)—the conscious ability to induce goosebumps—whose physiological study, to our knowledge, is confined to three single-individual case studies. Very little is known about the physiological nature and emotional correlates of this ability. The current manuscript assesses physiological, emotional, and personality phenomena associated with VGP in a sample of thirty-two individuals. Physiological descriptions obtained from the sample are consistent with previous reports, including stereotypical patterns of sensation and action. Most participants also reported that their VGP accompanies psychological states associated with affective states (e.g., awe) and experience (e.g., listening to music), and higher than typical openness to new experiences. These preliminary findings suggest that this rare and unusual physiological ability interacts with emotional and personality factors, and thus merits further study.

## INTRODUCTION

*"Certainly that heart is steel-framed which, in spite of one's chanting the holy name of the Lord with concentration, does not change when ecstasy takes place, tears fill the eyes and the hairs stand on end."*

Bhāgavata Purāna, 2.3.24, 800-1000AD (*Bhaktivedanta VedaBase, 2018*).

The study of exceptional individuals is central to the history of human neuroscience. Famous case studies include patients with specific neurological insults, such as naturally occurring lesions (e.g., Louis Leborgne; *Broca, 1861*), victims of crimes or accidents (e.g., Phineas Gage; *Harlow, 1848*), or patients receiving neurosurgery (e.g., Henry Melaison; *Corkin, 1984*). Detailed observations of these individuals were critical to initial developments within language processing, behavioural inhibition, and amnesia/memory consolidation respectively. Non-invasive imaging techniques allow contemporary research to pursue similar case study or small-sample approaches without autopsy, and relevant

Corresponding author
James A.J. Heathers,
jamesheathers@gmail.com

patients continue to make important contributions to the neuroscience of behaviour. For instance, patients B.G. and A.M., who have bilateral amygdalae lesions, show a differential panic response after a panic induction procedure, demonstrating that the amyagdala may not be necessary for the induction of panic (*Khalsa et al., 2016*). Inspired by this tradition, the present investigation seeks to identify, describe, and compare a small sample of people with atypical autonomic physiology who are otherwise healthy to identify corresponding patterns in their physical, cognitive, and emotional experiences.

An example here is illustrative: suppose two people, one healthy and neurotypical, the other identical but with a specific injury to the thoracic sympathetic ganglia. If the first walks face-first into a spider's web in the dark, we might expect subsequent reflex reactions (e.g., eye-blink, self-defensive posture) followed by the autonomic nervous system (ANS) increasing heart rate (HR) in preparation for whole-body mobilisation. The second would most likely have typical reflexes, including rapid appraisal of the situation (i.e., a small but definite cutaneous sensation on the face), and then subsequently a normal cognitive response (i.e., recognition of the presence of a spider, generally categorized by humans as dangerous and/or disgusting), but *reduced physical symptoms of fear or panic*, as their heart remains steady, palms dry, and so on. That is, this 'sympathetically-impaired' person would experience a dissociation between his or her appraisal of what is typically fear-inducing and the typical embodied experience of fear. If we consider emotion an interface between cognitive interpretations and physical experience of the body (e.g., *Schachter & Singer, 1962*; *Damasio, 1999*; *Barrett, 2017*), how should we understand perception of the relevant physiology being altered? What then would the experience be?

As an additional layer of complication, consider the above scenario in an individual with strong conscious control over their HR. How does one experience an embodied mental state when he or she can consciously control its autonomic precedents or antecedents? Somewhat paradoxically, the relevant control in this case would be over the ANS, the portion of the peripheral nervous system traditionally defined as operating outside volitional or conscious control, from the Greek *autos* (self) and *nomos* (law), i.e., self-governing. However, delineating autonomic and somatic (i.e., voluntary) functions is not clear-cut. Respiration is the classic example of an autonomic mechanism directly amenable to conscious control. Absent any form of conditioning, there is also evidence of volitional control of HR (*Bell & Schwartz, 1975*; *White, Holmes & Bennett, 1977*) and blood pressure (*Lowdon, Murray & Langley, 2011*). As autonomic systems co-modify, the above effects may be combined. For instance, conscious control of respiration might also be considered a deliberate modification of blood pressure (*Joseph et al., 2005*). Some curious case studies from meditative traditions concerning HR control (*Young & Taylor, 1998*) and temperature (*Benson et al., 1982*) have also been reported. Related research has additionally taken place under the rubric of biofeedback, where conscious access to a previously non-conscious ability is strengthened by physiological instrumentation (e.g., somatic control of skin temperature; *Taub & School, 1978*). Finally, a separate case again is the conscious control
**Table 1 Characteristics of individuals displaying voluntarily generated piloerection (VGP).**

| Reference | Age | Age discovered | Sites | Response time to completion | Response time to decay |
|---|---|---|---|---|---|
| Maxwell (1902) | 27 | 11–12 | Hips, thighs, back, arms | 2–10 s | 2–10 s |
| Lindsley & Sassaman (1938) | "middle aged" | 10 | Forearm, thigh, lower leg | 7 s | 15–20 s |
| Benedek et al. (2010) | 35 | 27 | Neck, spine, arms | 5.15 s | 10.8 s |

of pupillary contraction and dilation, which is both capable of being conditioned (*Cason, 1922*) and responds directionally to imagined changes in luminance (*Laeng & Sulutvedt, 2014*).

In addition to the above, there is an autonomic mechanism yet to be systematically investigated in terms of volitional control –*piloerection*, often referred to as 'goosebumps.' Involuntary piloerection is commonly observed during cold, fearsome, or intensely evocative emotional stimuli, and is subserved by the pilomotor projections of the sympathetic nervous system (SNS). However, while evidence in the scientific literature suggesting piloerection *can* be consciously controlled has been observed in only three published case studies over a period of more than a century (*Benedek et al., 2010*; *Lindsley & Sassaman, 1938*; *Maxwell, 1902*), the incidence of this ability, method of action, and potential psychological correlates have not been thoroughly examined. Voluntarily generated piloerection (VGP) is described below in brief (Table 1).

In light of other examples of conscious physiological control, VGP is unusual for several reasons. First, the *arrector pili* which control erection of individual hairs are smooth muscles—they have no somatic control in the manner of skeletal muscles, such as the biceps or quadriceps. Rather, they operate outside of conscious awareness, contracting or relaxing to local cellular factors or the general neurochemical environment. A reductionist physiological account would conclude conscious piloerection is not possible. Second, VGP was described in all three prior case studies as being 'discovered' by the person experiencing it. That is, they all outline various experiences where VGP was uncovered as a latent ability without conditioning, biofeedback, expectation, or training. Thus, no pathway to acquire this ability is known. Third, VGP offers control over a physiological phenomenon intimately involved in emotional experience, wherein bodily sensations, emotional states, and experiential terminology frequently overlap ('her palms were damp,' 'he had butterflies in his stomach,' and 'their hearts were pounding'), and likewise here 'my hair was standing on end.'

Both classic and modern accounts of the nature of emotion consider the integrated role of perception and feedback of somatic symptoms such as goosebumps (e.g., *Schachter & Singer, 1962*; *Damasio, 1999*; *Barrett, 2017*). Other self-perceived physical symptoms, such as the ability to feel the heart beating in the chest ('cardiac interoception'), associate with emotional (*Pollatos, Kirsch & Schandry, 2005*) and physical (*Herbert, Ulbrich & Schandry, 2007*) self-regulation. Involuntary piloerection has primarily been studied in the context of intense emotional experiences, has been shown to correlate strongly with reported emotional intensity, and seems to be particularly associated with states of awe (*Schurtz*

*et al., 2012*) or being 'moved' or 'touched' (*Benedek & Kaernbach, 2011*; *Wassiliwizky et al., 2015*). Whether VGP relates in any way to subjective experiences associated with involuntary piloerection remains an open question. Likewise, relationships between VGP and the tendency to experience *involuntary* piloerection is unknown. Finally, very little is known about psychological correlates of VGP. Is it accompanied by emotions and sensations usually associated with involuntary piloerection? Do people capable of VGP use this ability to enhance or moderate their psychological experiences?

Only one of the three case studies cited above mentions any psychological aspect; the participant in question reported that voluntary piloerection was experienced as "rather pleasurable than otherwise" (*Maxwell, 1902*, p. 373). This individual further reported unusual features—he could use VGP for headache relief, observed no special sensitivity during fear, sympathy, or music, and experienced involuntary piloerection most of the time while watching football (during which VGP was reportedly unusually easy). The other case studies primarily focus on automated detection or physiology of VGP (*Benedek et al., 2010*; *Lindsley & Sassaman, 1938*).

Another question of interest is whether people capable of VGP have unique personality profiles. *Maxwell (1902)* initially speculated that the ability may associate with a neurotic personality disposition. Since then, individual differences have been found in the propensity to experience involuntary piloerection and related psychological states and are most consistently associated with the personality domain of openness to experience—a domain reflecting variability in cognitive exploration (*DeYoung, 2015*). A questionnaire item reporting on experiences of aesthetic chills in response to music and poetry is one of the highest loading items on the openness to experience questionnaire, suggesting that experiences of aesthetic chills are a universal marker for openness to experience (*McCrae, 2007*). Furthermore, openness associates with chills in response to music (*Colver & El-Alayli, 2016*; *Nusbaum & Silvia, 2011*; *Silvia et al., 2015*), and the arts more generally (*Silvia & Nusbaum, 2011*). Open people are also found to be more prone to being absorbed in their experiences generally (*Wild, Kuiken & Schopflocher, 1995*), and in aesthetic contexts specifically (*Silvia & Nusbaum, 2011*). They are also more likely to report experiences of awe (*Shiota, Keltner & John, 2006*), and recall awe-based crying experiences with music (*Cotter, Silvia & Fayn, 2018*). Given the lack of consistent relationships between involuntary piloerection or chills and other personality domains, no hypotheses were formulated for other personality domains. However, given that individual differences in VGP have never been previously explored, a short measure of the Big Five was administered for exploratory comparisons with normative data.

Much of this prior work addresses the relationship between openness and experience of 'chills.' While chills are not identical to involuntary piloerection in the sense that they may not necessary involve either body hair moving visibly or the skin dimpling due to activity of the *arrector pili*, and they have distinct physiological correlates (*Sumpf, Jentschke & Koelsch, 2015*; *Wassiliwizky et al., 2017b*) experiences of chills and involuntary piloerection occur in response to similar situations (music, poetry, art) and in concordance with similar emotional experiences (awe, being moved). Therefore, we hypothesize that openness

associates with a greater propensity to experience both chills and involuntary piloerection, though to our knowledge, no study has previously explored this possibility.

The links between involuntary piloerection, the experience of chills, and existing personality correlates suggest that individuals with voluntary control over these symptoms tend to experience states of being moved, touched, and/or awed more frequently, and score higher on openness to experience measures. However, hypothesised differences in openness are contingent on whether the experience of VGP is accompanied by psychological experiences associated with involuntary piloerection. In summary, VGP provides potential insight into both the nature of autonomic regulation and a correlate of emotion and personality. The present study is the first we are aware of to characterise this ability in a sample consisting of more than one individual.

## METHOD

### Participants

Participants in the study were concurrently recruited from two distinct pools and all provided consent via a digital consent form.

#### Online sample

Given the suspected rarity of VGP, the first pool was recruited online via advertisements placed on psychology-relevant Facebook groups, and requests placed on any website mentioning congruent phenomenon (i.e., any full or partial description of a provocation of goosebumps under volitional control). All links provided referred to our own Facebook group created for recruitment (https://www.facebook.com/voluntarygoosebumps/), which gave a full general description of the study before linking participants to the questionnaire, in order to reduce confirmation bias.

Thirty-five participants completed some of the survey, but only thirty-two ($n = 32$; 22 males) answered all physiology-related questions on which exclusion criterial were based, two people did not complete the personality questions ($n = 30$), and an additional two others did not complete the aesthetic experiences scale ($n = 28$). Missing data was not imputed; relevant indices were calculated with cell sizes of existing values. Participants were excluded only from analyses that involved variables with missing data.

#### Mass undergraduate screening

The second pool of participants consisted of first year psychology students recruited via a mass screening questionnaire ($n = 682$). For the screening, the following question was asked:

> "Goosebumps are raised hair follicles that appear on your arms and other body parts when you are cold, afraid, or experience heightened emotions. However, some people can give themselves goosebumps just by thinking it. Is this something you can do?"

Endorsement of the screening question provided an indication of VGP prevalence in a large student sample. Those who endorsed the screening question above ($n = 120$) were invited to participate in the study, twenty-two completed the survey as above (i.e.,
physiology, personality, and emotional experience scales). To ensure that the final sample consisted of individuals who genuinely have the ability for VGP, this data remained un-analyzed until we had set criteria, as described below, based on previous case studies and results of the general web survey.

## Procedure

Participants from both recruitment pools completed the study online via a series of questionnaires programed in Qualtrics. After reading a general description of the study participants were subsequently asked for consent, after which all questionnaire measures were delivered. The study was approved by the ethics committee at Sydney University (Project #2015/598).

## Measures

The questionnaires administered consisted of questions regarding personal demographics (age/gender), physiological aspects of VGP, some standardized personality questionnaires with a particular focus on openness to experience scales, tendency to experience powerful aesthetic emotions, and experiential/psychological questions.[1]

[1]See Supplemental Information for full questionnaire.

### Physiological questionnaire

This questionnaire was modeled off the original case studies, and assessed age of discovery of VGP, important bodily sites (where the sensation initiated, where it occurred, and where it was strongest), response latency and decay, any control over body parts or sidedness, relationship with the respiratory cycle, and any potential practice schedules. Participants were also asked to describe in detail their skill, and the precise series of events used to precipitate VGP.

### Personality

*Big five personality.* Personality was assessed via the Ten Item Personality Inventory (TIPI; *Gosling, Rentfrow & Swann, 2003*). The TIPI is a brief assessment of five personality domains that has been administered extremely widely (i.e., to more than 300,000 people), and thus has well-established norms (Norms for the Ten Item Personality Inventory; SD Gosling, PJ Rentfrow & J Potter, 2014, unpublished data). The five domains are assessed with two items each, thus 10 items in total, with a 7-point Likert scale ranging from 1 (Disagree strongly to 7 (Agree strongly). Such short measures tend to suffer from low reliability—this is an acknowledged feature of personality domains measured with few items, and with this scale in particular (*Credé et al., 2012*; *Gosling, Rentfrow & Swann, 2003*). In the current sample, particularly low reliability was observed for openness (Cronbach's $\alpha = 0.26$) and emotional stability domains ($\alpha = 0.37$), with slightly higher consistency for agreeableness ($\alpha = 0.55$), conscientiousness ($\alpha = 0.62$), and extraversion ($\alpha = 0.65$) domains. Given the low reliability of these scales, we compared our VGP sample to population norms on both two-item domains and individual items.

*Openness/Intellect domain.* We also administered a subscale of the Big Five Aspect Scale (BFAS) to assess the openness/intellect domain more thoroughly. The BFAS divides each personality domain into two aspects (*DeYoung, Quilty & Peterson, 2007*). In the case

of openness, aspects reflecting sensory exploration (openness) are differentiated from abstract/semantic exploration (intellect) (*DeYoung, 2015*). Each aspect is assessed with 10 items (20 items total), with a 5-point Likert scale ranging from 1 (Disagree strongly) to 5 (Agree strongly). In the current sample, internal consistencies for the overall domain ($\alpha = 0.84$), as well as openness ($\alpha = .68$) and intellect ($\alpha = .88$) aspects were acceptable. No other subscales from the BFAS were administered.

*Aesthetic experience scale.* To test whether VGP individuals in our sample were more likely to experience strong aesthetic states, we administered the aesthetic experience scale (*Silvia & Nusbaum, 2011*). This scale was developed to assess individual differences in the frequency of powerful aesthetic experiences associated with involuntary piloerection. The scale breaks up into three factors that assess chills (three items), feeling touched (two items), and absorption[2] (five items). Participants responded to each item using a 1 (Never or Rarely) to 7 (Nearly Always) scale. In the current sample, internal consistency was good for all three scales ($\alpha = 0.72 - 0.89$).

*Other psychological experience questions.* To assess whether VGP associates with psychological sensations, we asked participants whether VGP accompanied experiences with involuntary piloerection. We used items from the aesthetic experience scale (excluding the chills items), where participants could choose any number of experiences (feelings of awe and wonder, feeling touched, feeling like crying, feeling absorbed or immersed, losing track of time, feeling like you are somewhere else, and feeling detached from your surroundings) accompanying VGP.

We also asked participants whether they used VGP to produce goosebumps during different activities such as listening to music, watching film or TV series, viewing art, reading literature or poetry, engaging in creative activities, watching dance, theater, playing computer games, and an 'other' option where participants could fill in a free response indicating the activity during which they use their ability. Finally, we included a yes/no item asking participants whether they used their ability to prolong involuntary piloerection.

## RESULTS

### Online sample

When asked to describe the process of VGP, thirty-two out of thirty-five participants in the online sample provided a detailed description. These open-field responses overwhelmingly described a process which was physical and reflex-like, rather an exercise of the imagination or re-experiencing:

P3:     *"Decide I want to give myself goosebumps, and with my next exhale, they come..."*

P4:     *"I think about goosebumps, they start to appear, I shudder/shiver, and there they are."*

P10:    *"I tighten a muscle behind my ears … and the goosebumps appear on my back and then travel to my arms."*

P11:    *"I just concentrate on the back of my neck and I get them."*

[2]While this state is labelled in the same way as the trait construct of absorption, within this manuscript absorption is considered a strong engagement state that has strong associations with experiences of awe and immersion.

| P12 | *"I tense my ears and scalp, (which also feels like i'm tensing the inside of my head) which sets off a shiver that runs down the back of my neck and around to my arms and chest."* |
| P18 | *"When I intend to cause them - outside of meditation (i.e; willfully), I focus conscious attention above and towards the back of the neck/base of skull and allow it."* |
| P20 | *"When you hear of news that's life-changing, you may gasp, open your eyes wide, and suddenly feel very cold. When I mimic these physical reactions...I can reproduce the effect coming from the back of my neck."* |
| P21 | *"I have to be fairly relaxed and not distracted, but I basically just think about the feeling itself and that is usually enough."* |
| P24 | *"I simple [sic] think of doing it. I don't need to have a [sic] emotion involved, in fact can do it now without feeling any emotion whatsoever."* |
| P25 | *"I just think about the act of having goosebumps and focus a little on the nape of my neck."* |
| P28 | *"I think of this energy (I have no name for it) and then it comes."* |
| P32 | *"I flex a "muscle"...in my brain. Sometimes i have to concentrate a little if i've been doing it for a while."* |

## Demographic and physiological features of VGP

Online sample participants who met the above criteria were typically adult (mean age = 32.44 yrs.; SD = 10.55) and discovered their VGP ability as teenagers or young adults (mean age of discovery = 16.91 yrs.; SD = 6.17). Most had substantial experience with the ability (15.53 yrs.; SD = 10.75). Only two participants reported discovering VGP as a latent ability when they encountered the study recruitment material. Location was broadly international, but largely from Anglophone countries ($n = 14$, USA; $n = 3$, UK, Australia; $n = 2$, New Zealand, Canada, Argentina; $n = 1$, Germany, Norway, Wales, Austria).

Participants reported the sensation as beginning on the back of the head/neck (71.88%) far more than any other site (e.g., arms, 21.86%; top or sides of head, 15.63%). Every participant reported the physical sensation of goosebumps on multiple bodily sites (mean = 5.75), overwhelmingly on the arms (90.63%), but also commonly on the back of the head/neck (75%), upper back (71.88%), shoulders (65.63%), legs (62.5%), top/sides of the head (53.13%), or lower back (46.88%). The sites where the sensation was reported strongest were the back of the head/neck (65.63%) or arms (53.13%).

Participants overwhelmingly reported the response as quick to initiate; 81.3% said the response began instantaneously (i.e., 0 to 1sec) or quickly (2 to 4sec). Time taken to initiate and then withdraw the response was highly variable, ranging from almost instantaneous to more than 15 s. In general, participants reported no control over the location of the goosebumps they provoked (75%), having them occur equally on both sides of the body (68.75%), and no control over 'sidedness' (71.88%). Participants generally reported being extremely or somewhat surprised that the general population does not share their ability (71.88%). They primarily described their experience of VGP as very easy or effortless (65.63%) or requiring 'some concentration or focus' (28.13%). Only two participants
(6.25%) described the phenomenon as 'effortful.' Very few participants described the experience of VGP as solely requiring imagination or emotion, such as recalling a moving, sad, or dangerous scenario (15.63%); the rest reported that VGP was possible as a purely physical phenomenon.

The respiratory cycle appeared to have no influence over the phenomenon, as participants reported being able to produce VGP during inspiration (81.25%), expiration (81.25%), with full lungs (78.13%), or with empty lungs (59.38%). Participants reported that they practised their ability using a variety of schedules: at least weekly (37.5%), at least monthly (15.63%), rarely (9.38%), or never (37.5%).

## Psychological features of VGP

Most participants (71.88%) reported at least one psychological state previously found to relate to involuntary piloerection accompanying their VGP experience. The most common reported were absorption/immersion (53.13%), awe/wonder (46.88%), and detachment (37.5%). The least commonly reported were feeling: 'touched,' 'being somewhere else,' 'crying,' or 'losing track of time.' The three most endorsed items are all part of the absorption/immersion subscale of the aesthetic experiences scale reflecting the propensity to experience powerful states of engagement and awe (*Silvia & Nusbaum, 2011*), suggesting that feelings accompanying VGP are similar to reports of awe-induced involuntary piloerection (*Schurtz et al., 2012*). Since much of the recent work on involuntary piloerection links the phenomenon to a state of being moved (*Wassiliwizky et al., 2017a*; *Wassiliwizky et al., 2017b*), one might expect our sample to also endorse items in the touched subscale of the aesthetic experience scale, but this was not the case. From one perspective, this may indicate that experiences during VGP are distinct from experiences of being moved. On the other hand, it is difficult to imagine feelings of being moved in the absence of a stimulus that moves you, as is imagining being moved without being absorbed by the stimulus to some degree. Participants also reported the following normal cutaneous sensation of goosebumps during commonly piloerective stimuli: when cold, during music, during films or television, when afraid, when touched lightly, during the experience of aesthetic or natural beauty, etc. (note: these are in decreasing rank order).

Participants generally reported to have no personal explanation for their ability (37.5%) or offered a scientific mechanism (37.5%; descending noxious inhibitory control, various neurotransmitters, the autonomic nervous system, circulating epinephrine, biofeedback, etc.) Only a single participant mentioned anything akin to a spiritual or non-mechanistic process.

Next, we asked whether participants used their VGP during activities that tend to elicit involuntary piloerection. Of the sample, 71.88% reported using their ability in response to at least one of the activities. The most frequently reported activities where participants used their ability were music (53.13%), film/television (28.13%), literature (25%), creative activities (25%), art (21.88%), dance (18.75%), theatre (15.63%), and gaming (12.5%). The 'other' category was also frequently endorsed (46.88%), and included meditation,

engagement with nature, exercise, sex, work or study for added energy and concentration, and to warm up. Only one participant mentioned using the ability to alleviate pain, as was the case in *Maxwell (1902)*. Thus, most of the sample reported using their ability in contexts where involuntary piloerection is frequently reported. Around half (53.13%) reported deliberately using their ability to prolong experiences of involuntary piloerection.

To test whether VPG-capable individuals experience emotions and experiences associated with involuntary piloerection more frequently, we compared their scores on the aesthetic experiences scale to the largest available sample previously collected from university students ($n = 188$; *Silvia & Nusbaum, 2011*). VGP-capable participants ($M = 4.75$, $SD = 1.50$), compared to a previously collected sample ($M = 4.55$[3]), reported no difference in the frequency of feeling absorbed (mean difference $= .20$; $t(27) = 0.69$, $p = .50$, Cohen's d $= .13$, 95% CI $[-.39-.78]$), and no difference ($M = 3.98$, $SD = 1.67$ versus $M = 3.63$) in the frequency of chills (mean difference $= .26$; $t(29) = 1.14$, $p = .26$, Cohen's d $= .21$, 95% CI $[-.27, .97]$), and no difference ($M = 3.55$, $SD = 1.56$ versus $M = 4.13$) in being moved (mean difference $= -.58$; $t(29) = -2.02$, $p = .053$, Cohen's d $= .37$, 95% CI $[-1.16-.08]$).

Since we had no prior information regarding sample characteristics of our VGP-capable participants, we compared personality data to normative data from the largest published samples for the relevant scales (Table 2). The VGP sample was significantly higher on BFAS Openness/Intellect, in both the domain and aspect level, and the differences produced large effect sizes. This was not replicated with the TIPI measure of openness, but the difference was significant for one of the openness items. No other differences were observed at the domain level of the TIPI, but there were some differences at the item level. However, none met Bonferroni-corrected threshold for significance ($\alpha/10 = 0.005$). The VGP sample scored significantly lower on an agreeableness item, and significantly higher on an emotional stability item; both differences were medium in effect size.

As a post-hoc follow-up, Fig. 1 compares the BFAS Openness/Intellect omnibus score against large samples where relevant data was available (8 samples drawn from six papers; *Antinori, Carter & Smillie, 2017*; *DeYoung, Quilty & Peterson, 2007*; *Fayn et al., 2017*; *Fayn et al., 2015*; *Sun, Kaufman & Smillie, 2018*; *Weisberg, DeYoung & Hirsh, 2011*). Uncorrected $p$-values ($t$-test, two-tailed) range from 0.051 to $10^{-7}$, and effect sizes range from Cohen's $d = 0.35$ to 1.16.

## Mass undergraduate screening

From all of the above, we summarize representative VGP characteristics as follows; **(1) VGP occurs on a volitional basis**; participants report producing goosebumps via *an intentional and non-reflective* activity, as in a volitional movement; no exercise of the imagination (such as re-experiencing previous excitement or imagining trauma) is involved**; (2) VGP involves a specific pattern of sensation**; participants report a sensation *beginning at the back of the head/neck*, and *strongest or most noticeable on the back of the head/neck, or arms*; **(3) VGP involves low effort and short latency**; participants report the sensation as *beginning within a few seconds, and without strong effort.*

[3] Means and standard deviations for the subscales of absorption, chills, and touched were not reported in *Silvia & Nusbaum, (2011)*, therefore the means for these scales were calculated based on item means.

**Table 2  VGP sample vs. population norms for Ten-Item Personality Inventory items and domains, Big Five Aspect Scale subscales and domain.**

| Personality measure | Sample | | Norms | | Difference | | Confidence intervals |
|---|---|---|---|---|---|---|---|
| TIPI Items | Mean | SD | Mean | SD | p | Cohen's d | |
| Extraverted, enthusiastic. | 4.73 | 1.80 | 4.42 | 1.73 | 0.354 | 0.17 | [−0.19, 0.53] |
| Critical, quarrelsome (R). | 3.43 | 1.72 | 4.08 | 1.72 | 0.047* | −0.38 | [−0.75, 0.01 |
| Dependable, self-disciplined. | 5.50 | 1.36 | 5.00 | 1.54 | 0.055 | 0.37 | [−0.01, 0.73] |
| Anxious, easily upset (R). | 4.20 | 1.81 | 4.06 | 1.81 | 0.671 | 0.08 | [−0.28, 0.44] |
| Open to new experiences, complex. | 6.00 | 1.20 | 5.49 | 1.34 | 0.029* | 0.42 | [0.04, 0.79] |
| Reserved, quiet (R). | 4.10 | 1.58 | 3.54 | 1.88 | 0.063 | 0.35 | [−0.02, 0.72] |
| Sympathetic, warm. | 5.43 | 1.14 | 5.29 | 1.44 | 0.502 | 0.12 | [−0.24, 0.48] |
| Disorganized, careless (R). | 4.40 | 1.65 | 4.22 | 1.85 | 0.552 | 0.11 | [−0.25, 0.47] |
| Calm, emotionally stable. | 5.20 | 1.54 | 4.62 | 1.64 | 0.049* | 0.38 | [0, 0.74] |
| Conventional, uncreative (R). | 5.23 | 1.78 | 5.52 | 1.54 | 0.385 | −0.16 | [−0.52, 0.20] |
| | | | | | | | |
| TIPI Domain | | | | | | | |
| Extraversion | 4.42 | 1.46 | 3.98 | 1.59 | 0.114 | 0.30 | [−0.07, 0.66] |
| Agreeableness | 4.43 | 1.21 | 4.69 | 1.23 | 0.257 | −0.21 | [−0.57, 0.15] |
| Conscientiousness | 4.95 | 1.29 | 4.61 | 1.42 | 0.161 | 0.26 | [−0.10, 0.62] |
| Emotional Stability | 4.70 | 1.32 | 4.34 | 1.48 | 0.145 | 0.27 | [−0.09, 0.64] |
| Openness | 5.62 | 1.15 | 5.51 | 1.14 | 0.604 | 0.10 | [−0.26, 0.45] |
| | | | | | | | |
| BFAS | | | | | | | |
| Openness/Intellect | 4.05 | 0.48 | 3.60 | 0.51 | 0.00002* | 0.93 | [0.50, 1.36] |
| Openness | 4.03 | 0.50 | 3.68 | 0.61 | 0.0005* | 0.72 | [0.31, 1.11] |
| Intellect | 4.07 | 0.68 | 3.53 | 0.63 | 0.0001* | 0.8 | [0.39, 1.21] |

**Notes.**
 Reversed items are marked (R) and presented recoded in line with domain label.

After the three criteria above were determined, the undergraduate data collected from a local sample was opened and analysed. Per the criteria, none of the undergraduates surveyed reported a congruent VGP phenomenon, and a follow-up investigation could not be conducted.

## DISCUSSION

To our knowledge, this paper outlines the first sizable sample characterised who report having a voluntary ability to produce goosebumps. While this sample was not observed directly in a laboratory setting, participants principally reported an experience consistent with those previously identified in laboratory case-studies over the previous century (*Benedek et al., 2010*; *Lindsley & Sassaman, 1938*; *Maxwell, 1902*) and to each other.

As VGP is almost entirely unknown, even within the scientific study of autonomic physiology, we doubt that study participants were subject to demand characteristics. It is unlikely any participant had prior cues or expectations regarding how their ability might be expected to work due to its rarity. Participants also did not have access to each other's responses. In open-field descriptions, no participant reported any aberrant or unusual phenomena, such as goosebumps on glabrous skin areas, a fast or instantaneous

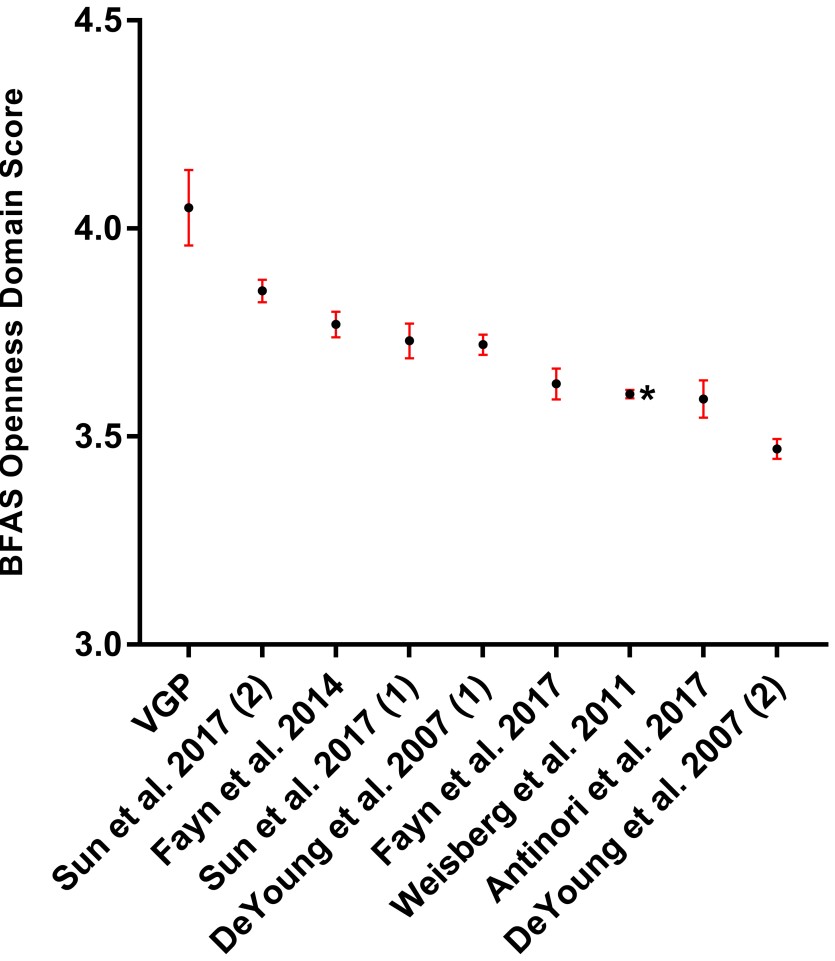

**Figure 1 VGP-capable sample score on BFAS Openness Domain vs. other available means.** Results are shown as mean ±SE. The estimate forming more than half of the available data points ($n = 2,643$; *Weisberg, DeYoung & Hirsh, 2011*) was taken as the population norm, and in the below is marked with an asterisk.

piloerection response (i.e., one where the sensation is quicker than the possible latency of the SNS), or an irregular dermatomal pattern. Neither were any parethesias described –itchiness, tingling, burning, and so on. Instead, the cutaneous component of VGP, its latency, and the body parts involved appear to be congruent with normal involuntary piloerection (i.e., identical to those experienced during cold, emotional elicitation, fear, etc.). No participant mentioned any injury, neurological insult, or pre-existing condition, relevant or otherwise, in their description of how VGP was possible. Overall, the responses we received paint a consistent picture of VGP as a perfectly normal and straightforward phenomenon, one which is curious, sometimes pleasant, and essentially benign.

Having established that VGP is reported as previously described, these results both confound our understanding of voluntary control over the autonomic nervous system and provide some insight into its phenomenon. *Lindsley & Sassaman (1938)* debated at length whether VGP was due to an innate skill or to some form of conditioning, where an original piloerective stimuli (cold, fear, etc.) acts as an earlier unconditioned stimulus. Conditioning seems unlikely to us for three primary reasons. First, two participants discovered they could perform VGP only upon receipt of the study materials. Second, many describe stumbling across VGP either purely out of idle self-experimentation, often at a young age, or sometimes in a deliberative attempt to re-create a previous experience. For an association to form, it would have to be entirely without conscious awareness. Third, if the conditioning hypothesis were true, it would result in a remarkably consistent phenomenon which centres around a deliberative point on the back of the head or neck. There is no physiological explanation for why or how localised focus or tension is the most common physical trigger for a conditioned response.

It is also difficult to integrate the above into classic observations of piloerection from animal models. Animals display piloerection as a functional social signal in both aggression and defense (as well as during cold exposure), it can be observed on localised and generalised bodily sites, and the response can be evoked by direct stimulation of a variety of sites within the brain and periphery (*Blanchard & Blanchard, 1977*; *Maickel et al., 1967*; *Shaikh, Barrett & Siegel, 1987*). However, in humans, we assume the ability is vestigial and observe little interindividual variation in site (except in the case of specific medical conditions such as pilomotor seizures; *Loddenkemper et al., 2004*).

With respect to psychological features of VGP, most participants reported that it associated with emotional states observed during involuntary piloerection, such as absorption/immersion, and awe. This suggests that VGP-capable individuals may possibly be able to voluntarily elicit these emotional states if provoking the physical experience has direct congruence with emotional elicitation. This finding is in line with research demonstrating greater emotional reactivity after artificial induction of piloerection (*Fukushima & Kajimoto, 2012*). Further, approximately 72% of the sample reported using their ability to produce piloerection during various aesthetic activities. Of particular interest is one of the unsolicited uses reported, namely for attention/concentration (work, study, testing situation, lecture). Given the link between attention and emotion in terms of selection, orientation, and engagement (see *Yiend, 2010* for a review), these attentional uses are congruent with reports of greater absorption and awe during VGP. Using VGP ability to control attention during situations that require it is intriguing, and a viable target for future confirmatory research.

While some differences were observed at the item level of the TIPI, none survived a correction for multiple comparisons. Thus, we leave the possibility of broad personality differences for future confirmatory research and will focus on discussing the strongest and most consistent finding—participants were noticeably higher than population norms on openness to experience. Unpacking these findings at this stage is speculative, but one possibility is that open people invest more time and effort into self-exploration. Openness is the strongest personality predictor of self-reflection (*Trapnell & Campbell, 1999*), and

indeed some participants reported the discovery of the ability through self-experimentation and deliberate attempts to re-recreate a previous experience. However, the direction of causality cannot be assumed.

The higher openness reported by our sample does not appear to be particularly driven by one aspect of openness to experience, but rather, our sample was higher in both openness and intellect. This distinction is theorised to reflect individual differences in cognitive exploration of sensory (openness) versus abstract (intellect) stimuli and situations (*DeYoung, 2015*), but also emotional versus cognitive engagement with sensory and abstract information (*Fayn et al., 2017*). More specifically, openness positively, while intellect negatively, predicts self-reported frequency of chills (*Silvia & Nusbaum, 2011*). Thus, differences in openness were expected. In fact, the openness scale includes items that assess tendencies for absorption (e.g., "Get deeply immersed in music," "Enjoy the beauty of nature") and thus one might expect a stronger relationship with openness compared to intellect; however, the VGP sample was higher on both. While circularity in measurement could be claimed for openness, there are no items within the intellect scale that have such overlap.

## FUTURE DIRECTIONS

Having established that VGP can be reliably identified in multiple individuals, the immediate extension of this work is to confirm the present findings under controlled laboratory conditions, and to add to previous observations. For instance, what overlap can be observed between other sympathetically-mediated sites in the periphery –does VGP imply a parallel response in the microvasculature, HR, or skin resistance? Moreover, does it augment or antagonise sympathetic reflex responses such as startle? Finally, if VGP is an analogue of an emotional response, does it alter the nature of emotional experience to stimuli (both naturalistic and experimental)?

These questions approach constructs of interest within broader theories of emotion; for instance, constructionist theories of emotion. As an example, the Conceptual Act Theory (CAT) understands emotion from a constructionist perspective, where they are synthesised from physically perceived representations combined with an understanding of emotional categories themselves (e.g., *Barrett, 2014*). Interestingly, CAT proposes that emotions are born from interoceptive information (perceived from inside the body; e.g., the sensation of the heart beating or stomach turning), exteroceptive information (taste, smell, sound, etc.), and their integration with language in a 'conceptual act.' In this context, VGP is unusual in the sense of being able to produce both an internal sensation (most participants describe an internal process which has a distinct activity or pathway) *and* a cutaneous sensation (the physical experience on the skin itself). A comparison to the experience of false haptic feedback, a manipulation where hairs on the skin are raised using electrostatic charge, without any internal experience (*Fukushima & Kajimoto, 2012*) would be instructive.

## LIMITATIONS

There are several limitations in the present study. First, as we failed to identify any participants from a large sample of local undergraduates, the investigation presented is purely descriptive by necessity—we had no opportunity to directly study the physiology of VGP. Second, it is impossible to establish how many times our initial recruitment materials or survey were viewed, and VGP is sufficiently rare that we could not identify one in a sample of several hundred participants drawn from a general population. Thus, we have no reasonable estimate of VGP base rate. Considering this, and the fact that it has never been previously outlined, we can conclude nothing more than that VGP is reasonably uncommon. Third, while the BFAS openness/intellect measure demonstrated acceptable reliability, the TIPI measure was mostly unreliable. This is not uncommon when trying to measure personality domains with two items per domain. While the TIPI has been shown to correlate strongly with broader measures of the Big Five, short measures have been shown to increase Type 1 and Type 2 error rates (*Credé et al., 2012*). Further, past work has questioned the validity of the openness items within the TIPI, suggesting that they fail to capture central elements of openness to experience (*Hofmans, Kuppens & Allik, 2008*), which may explain the inconsistency between the two different measures of openness in our sample. Fourth, self-selected participants accessed via Internet recruitment answering a survey presumably out of curiosity might be assumed to display more personal openness. To address this requires either a carefully matched sample to any remotely collected data or identifying a VGP-capable population who are not self-selected. As outlined above, this is a significant undertaking when studying a rare phenomenon.

## CONCLUSION

This study is the first we are aware of to outline self-reported ability to produce voluntary piloerection in a sample of more than a single individual. This phenomenon was consistent between individuals, reasonably effortless, visibly distinct, and we can tentatively conclude has correlates within the personality domain of openness. Participants who self-identified as being VGP-capable were generally aware of their ability, regard it as harmless or pleasant, and often use it to modify or prolong sensory experiences. Individuals who display VGP may play an important role within the future study of emotion and emotional regulation, as the role of the ANS integrated within the physiology and experience of visceral emotions (shock, awe, being moved, fear, panic, disgust, etc.) is potentially illuminated by individuals with rare or unusual physiology.

## ACKNOWLEDGEMENTS

The authors would like to thank the anonymous participants who volunteered to be involved in this study. We are immensely grateful for the insights you have provided.

### Funding

This work was supported by the Research Fund of KU Leuven (GOA/15/003; OT/11/031), and by the Interuniversity Attraction Poles program financed by the Belgian government (IAP/P7/06). The funders had no role in study design, data collection and analysis, decision to publish, or preparation of the manuscript.

### Grant Disclosures

The following grant information was disclosed by the authors:
KU Leuven: GOA/15/003, OT/11/031.
Belgian government: IAP/P7/06.

### Competing Interests

The authors declare there are no competing interests.

### Author Contributions

- James A.J. Heathers and Kirill Fayn conceived and designed the experiments, performed the experiments, analyzed the data, contributed reagents/materials/analysis tools, prepared figures and/or tables, authored or reviewed drafts of the paper, approved the final draft.
- Paul J. Silvia and Matthew S. Goodwin authored or reviewed drafts of the paper, approved the final draft.
- Niko Tiliopoulos contributed reagents/materials/analysis tools, authored or reviewed drafts of the paper, approved the final draft.

### Human Ethics

The following information was supplied relating to ethical approvals (i.e., approving body and any reference numbers):

The University of Sydney Human Research Ethics Committee granted Ethical approval to carry out the study (Project number: 2015/598).

### Data Availability

The raw data are provided in a Supplemental File.

### Supplemental Information

Supplemental information for this article can be found online at http://dx.doi.org/10.7717/peerj.5292#supplemental-information.

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
