# Peer review of "The voluntary control of piloerection"

_PeerJ, doi:10.7717/peerj.5292_

## Round 0.1 · original submission · Major Revisions

The reviewers have raised a number of issues that need to be addressed in the revised manuscript, in particular controlling the false discovery rate, appropriate interpretation of the results as self-report data, clarifying the use of two samples, and inclusion of relevant previous literature. The reviewers also made several suggestions to improve the text.

Reviewer 1 ·

Basic reporting

This is a manuscript on voluntarily generated piloerection (VGP) and its self-reprted physiological and psychological correlates. It is the first broad study to investigate this phenomenon. The draft is clearly written and well structured. Please find corrections and suggestions for change below.

1) Although clearly written, the text contains several unexpected turns which are not easy to follow. For instance, after describing introspective abilities and physical self-regulation (lines 113-116) the authors write: “In this tradition, the presence of piloerection during intense experiences has been shown to correlate strongly with reported emotional intensity […] (Benedek & Kaernbach, 2011, Wassiliwizky, Wagner, Jacobsen, & Menninghaus, 2015)…” The cited studies represent work on emotional induction using aesthetic stimuli. It is not clear why this research is in the tradition of the aforementioned work. Similarly, after reviewing a study on drug administration (lines 151-154) the authors proceed: “No hypotheses were formulated for other personality domains”, which doesn’t seem to be related to the previous.

2) Several relevant studies on piloerection are not mentioned or discussed in the manuscript. The authors state “there is an autonomic mechanism that to our knowledge has never been systematically investigated – piloerection”. However, several studies have done that, using objective approaches to measures piloerection: Craig (2005); Benedek & Kaernbach (2011); Kim, Seo, Cho (2014); Sumpf, Jentschke, Koelsch (2015); Wassiliwizky, Koelsch, Wagner, Jacobsen, Menninghaus (2017); Wassiliwizky, Jacobsen, Heinrich, Schneiderbauer, Menninghaus (2017). See also Schubert, Zickfeld, Seibt, Fiske (2016) for the link between goosebumps, tears and being moved and Nishida (1997) for animal piloerection in courtship behavior (missing in lines 419 et sqq: “Animals display piloerection as a functional social signal in both aggression and defence”).

3) In lines 148-151 the authors discuss the similarity between subjectively felt chills and objectively measured piloerection and conclude: “we may assume they are strongly related”. Two studies have looked into this issue empirically and come to different conclusions: Sumpf, Jentschke, Koelsch (2015) and Wassiliwizky, Koelsch, Wagner, Jacobsen, Menninghaus (2017).

4) Could the authors specify every time what kind of piloerection they are referring to? E.g. in line 116: “the presence of piloerection during intense experiences” should read “the presence of involuntary piloerection during intense experiences” or line 128-130 “he […] experienced [involuntary?] piloerection most of the time while watching football.”

Experimental design

no comment

Validity of the findings

5) I find it particularly problematic that the student sample was excluded from further analyses because the participants did not comply with the VGP-criteria that were deduced from the online sample. Why are the answers from the online sample given more weight? One could argue that the online sample does not fulfill the criteria set by the student sample.

6) The authors do seemingly not correct for multiple comparisons (see lines 349-352 or line 367). This can be problematic in terms of alpha inflation. Either the p values have to be corrected for multiple testing or the authors could run a multivariate omnibus test instead of several t-tests.

7) The results show that people capable of VGP mostly associate it with psychological states of absorption (53.13%) and least with “being touched” and “like crying”. In line 326 the authors state “VGP is associated with feelings congruent to experiences of involuntary piloerection”. However, I am not aware of any previous empirical work that links piloerection with absorption. On the contrary, several papers show a link to “being moved/touched” (Konecni, 2005/2007/2008; Benedek & Kaernbach, 2011, Wassiliwizky, Wagner, Jacobsen, Menninghaus, 2015; Seibt, Schubert, Zickfeld, Fiske, 2017). Taken together, this actually points in the direction of some differences between VGP and involuntary piloerection (rather than congruency).

8) The authors claim several times that VGP is an extremely rare phenomenon. However, in their study (pool 2) 120 out of 682 participants indicated to be able to self-induce VGP, which is about 18% of the sample!

Additional comments

minor points:

- Conceptual issue: One aspect in the introductory illustration (starting line 54) is not intuitive. The authors write “The second would most likely have normal reflexes […] and then subsequently a normal fear reaction […], but […] their heart remains steady, palms dry, and so on.” In most psychological accounts, the bodily component represents an essential part of the emotion. How can someone have a “normal fear reaction” without any physiological arousal? It is also surprising that the authors call the Schachter & Singer conceptualization of emotions (from 1962) a “modern account” (line 111).

- line 72: “nomos” is a singular form, please change the translation to “law”

- please explain what you mean by “sample of any size” (line 164) or change the wording.

- line 335: “we tested whether participants used their VGP abilities”; the more appropriate wording would be “we asked whether”

Reviewer 2 ·

Basic reporting

Generally the reporting is fine except for a few issues raised below.

Experimental design

Generally the methods are fine except for a few issues raised below. The primary concern is that the authors overstate the meaning and implications of self-report results.

Validity of the findings

My main concern is the use of self-report measures. The authors need to reduce the strength of the claims they make.

Additional comments

1. Tables
Table 1: Resp. time is unclear – I recommend abbreviating RT or spelling out Response.

Table 2: populations should be population, “and domain” is confusing here, consider rewriting for greater clarity. Confidence interval should be intervals. Define LL and UL – it seems these aren’t necessary.

2. Abstract
“physiological study in scientific history” seems a bit grandiose. I recommend just writing that only 3 previous cases have been reported to the best of your knowledge (this saves you from embarrassment in case some other cases are found …)

Rather than referring them as being “capable of VGP”, perhaps “report being capable of VGP”. It might also be valuable to state that their ability for VGP was formally documented/observed.

3. Introduction
Generally, I think the introduction could benefit from being more concise and having a clearer linear structure. “Non-invasive imaging …” – this sentence seems unnecessary. The second paragraph doesn’t seem to me valuable in the context of the introduction. The third paragraph seems unnecessarily long and there is repetition (eg. control of HR). The structure of the 4th paragraph needs to be edited to improve the flow. It seems strange to state that VGP provides insights into personality when you have yet to report any personality differences in those reporting VGP.

4. Method
“exclusion criterial was based” should be “exclusion criteria were based”

“Those who endorsed the question (n=120) were invited to participate in the study, twenty-two completed the survey”
What survey? This is unclear.

Qualtrix is misspelled.

The fact that you have 2 samples that were administered questionnaires is confusing. Please make this clearer in terms of what was done with each sample. How was the online sample contacted? How many participants started the survey and how many completed it? This should be clearer and an attrition rate should be clearly stated.

The low internal consistency for every subscale (the cut-off is typically .7) is a huge concern and thus I think it probably makes the most sense to ignore these data.

Please report the number of items and scoring format for all scales.

Results
It seems “Online sample” would be a more suitable label than “Pool 1”.

P18: willfully is a word and thus [sic] isn’t required.

Please report ranges and/or CIs for %s in the text.

The descriptions of these different features of VGP should include details regarding the range of the scale. Otherwise, it’s unclear (e.g. the question about effort).

How was was practice defined here? Participants who did the act regularly (because it’s interesting/enjoyable) may be classified as practicing even though they weren’t actually “practicing” per se. Please clarify.

The section at the beginning of Pool 2 (see my comment above), shouldn’t include bullet points – put this in a paragraph. Also, this looks more like a discussion so why not put it in the Discussion?

How do you know it’s not involuntary and they just incorrectly perceive it as voluntary? The fact that they don’t have complete control over it (eg side of body) and that it occurs under similar states as involuntary piloerection lends plausibility to this.

The absence of a control sample is a concern and should be addressed. The same goes for the self-selected nature of this sample.

Did you ask about frequency of involuntary piloerection? A comparison with a control group would be very valuable here.

Please report descriptive statistics associated with inferential statistics (eg Aesthetic experiences questionnaire).

I recommend having subsections in the text.

Please report effect sizes for all effects (eg Fig 1).

Have the authors corrected for multiple analyses? There are many analyses reported in Table 1; accordingly, I’m nervous about the “calm” effect (p=.049). If no correction was applied, I suggest applying one, such as false discovery rate correction.

Discussion
“sizable sample ever characterised who have a voluntary ability to produce goosebumps.”
“Having established that VGP exists as previously described …”

Insofar as none of this was physiologically corroborated and it’s all self-report, I think these and other sentences absolutely need to be greatly tempered. For the record, I believe this is a real phenomenon but I don’t find the evidence presented (all self-reports, questionnaires with poor reliability, all online participants, small sample size) to be particularly compelling.

“Third, if the conditioning hypothesis were true, it would result in a remarkably consistent phenomenon which centres around a deliberative point on the back of the head or neck.”
Please explain this – I don’t understand the logic.

“However, in humans, we assume the ability is vestigial and observe little interindividual variation in site (except in the case of specific medical conditions such as pilomotor seizures; Loddenkemper et al., 2004).”
Awkward phrasing, especially since the research team didn’t seem to author that paper. I don’t understand what you mean about variation in site here.

“This suggests that VGP-capable individuals may be able to voluntary elicit these emotional states.”
This is tricky as the authors (to my knowledge) didn’t probe how it was done. Were these individuals directly producing piloerection or were they producing affective states (eg awe) that trigger piloerection?

Absorption isn’t an emotion.

I would highlight that absorption is a predictor for a range of phenomena from anomalous experiences, drug-induced synaesthesia, hypnosis, mental health worries, etc.
https://www.ncbi.nlm.nih.gov/pubmed/29452127
https://www.ncbi.nlm.nih.gov/pubmed/28174078
https://www.ncbi.nlm.nih.gov/pubmed/28758796
https://www.ncbi.nlm.nih.gov/pubmed/24146659
https://www.ncbi.nlm.nih.gov/pubmed/15849873
It may be that individuals high in absorption have liminal boundaries for anomalous experiences and increased liminality confers greater control (as is the case with highly suggestible individuals in the context of hypnosis, for example) or at least the perception of control.

I would qualify the openness findings on the basis of the poor internal consistency of the measure.

Future directions
I think the first goal is to confirm this phenomenon in the lab under controlled conditions. I’m surprised that’s not listed first. Frankly, I think the authors are jumping a bit ahead.

Conclusion
“This study is the first to outline the ability to produce voluntary piloerection in a sample of more than a single individual.”
This is overstated – you don’t demonstrate this ability. You identify a small group of individuals who claim to have this ability. The same goes for all the characteristics (e.g., that it’s effortless) – it should be changed to clearly indicate that these individuals report these characteristics rather than that they are characteristics of this phenomenon. The same goes for the term “VGP-capable participants “

·

Basic reporting

This is an exploratory study investigating voluntarily generated piloerection. This is a rare and under-investigated phenomenon and this paper makes an important contribution toward better understanding both the characteristics of the phenomenon itself and traits that distinguish individuals capable of VGP. The article is clear and well written. The introduction successfully establishes the potential value of research in this area, and the findings and limitations of this work are fairly evaluated.

Experimental design

This is very much an exploratory study that focuses mainly on describing the features of VGP. The main formal analysis is comparison of a self identified sample of individuals capable of VGP with previously reported samples from past literature.
My main concern with this paper is the comparison of personality measures between pool 1 and previous data. I do not think it is appropriate to compare this data at both the level of domains and also individual items. In particular, conducting separate analyses for each individual item without correcting for multiple comparisons seems problematic. I note that line 439 acknowledges that reported differences at the item level would not survive correction. With this in mind I suggest omitting the item level analysis completely.

Validity of the findings

This study investigates a research area where almost nothing is known. As such the findings here, although primarily descriptive in nature are a valuable and important contribution to the literature.

Additional comments

There are no other major issues that need to be addressed in this paper. Below are some general comments and requests for further detail that the authors may optionally like to respond to.

- The first paragraph provides some examples of famous case studies that have progressed human neuroscience. There is also mention of two case studies (BG and AM) using non-invasive techniques, but there is no information provided as to the details of why these particular cases were important. I wonder if it would be possible to provide some very brief comments along these lines to more fully illustrate the specific ways that studying exceptional individuals can increase scientific understanding.

- Many of the references for studies showing control of processes that are typically considered autonomic are quite dated (lines 68-87). Is there a reason that recent research has not addressed these phenomena? Is this worthy of comment?

- Regarding lines 101-103, what does it mean to say that the arrector pilli are ‘smooth muscles’ without somatic control? For someone with limited knowledge of physiology this seems confusing. What is the property of these specific muscles that implies that conscious control should not be possible? More detail here would help.

- Line 176: what are the congruent phenomena that were targeted as part of recruitment?

-Line 199: Qualtrics

---

## Round 0.2 · accepted · Accept

As confirmed by the re-reviews, the authors adequately addressed all previous comments.

# Reviewer 1 ·

Basic reporting

Due to the changes the manuscript greatly improved in clarity. I have no amendments to make.

Experimental design

The authors now deal explicitly with the problem of alpha inflation and other issues I brought up in my last review.

Validity of the findings

All standards of the journal are met.

Reviewer 2 ·

Basic reporting

I have no further issues here.

Experimental design

I have highlighted my main concerns in my previous review.

Validity of the findings

As long as the authors highlight the central limitations (self-report measures, small sample sizes, online samples, some measures with poor consistency, multiple comparisons problem) and temper any strong conclusions, I think the results are fine to present as an exploratory study.

Additional comments

I have no further comments.

·

Basic reporting

In my view the original submission already fulfilled the requirements of basic reporting, and this revision has further increased the clarity and readability of the manuscript.

Experimental design

I previously expressed some concerns about potential duplication of analyses but I find the authors justification reasonable and accept that they have taken a sensible exploratory approach.

Validity of the findings

The findings are valid, interesting and well explained.

Additional comments

This is an impressive revision that appears to address all of the concerns raised. The added details, particularly in response to comments from Reviewer #1, have provided helpful additional context and situated these findings nicely within previously literature.
I congratulate the authors on a creative and engaging treatment of a puzzling and most interesting phenomenon.